# Can "sponge city" pilots enhance ecological livability: Evidence from China

**Qi Wang[1], Qinmei Wang[1], Xi Wang[ID][2]***

**1** International Business School, Shaanxi Normal University, Xi'an, 710100, China, **2** College of Water Resources and Environment, University of Jinan, Jinan, 250000, China

* kier_wong@163.com

## Abstract

The challenges posed by environmental pollution, water scarcity, and energy limitations resulting from industrialization and modernization pose significant threats to human habitats. Consequently, assessing ecological livability and delineating pathways for improvement carry considerable practical importance. Leveraging panel data encompassing 288 cities in China from 2010 to 2021, this study establishes an evaluation system for ecological livability, encompassing three dimensions: natural greenery level, residential comfort level, and environmental governance level. Subsequently, the study measures the ecological livability level and investigates the impact of "sponge city" pilots on ecological livability and their underlying mechanisms using a multi-period difference-in-differences model. Our findings underscore the substantial role of "sponge city" pilot projects in bolstering ecological livability, with robustness observed across various models and specifications. Specifically, human capital concentration and green technology innovation emerge as pivotal pathways through which "sponge city" pilots augment ecological livability. Moreover, the effectiveness of "sponge city" pilots varies across regions due to disparities in drought severity and water supply, with more pronounced effects observed in arid areas and cities facing water supply shortages. This research furnishes comprehensive theoretical and empirical underpinnings for comprehending the influence of "sponge city" pilots on ecological livability, offering valuable insights and recommendations to inform future efforts aimed at enhancing ecological livability and fostering sustainable development.

## Introduction

With the development of the world economy, population growth, and the acceleration of urbanization, the Earth's ecological environment faces immense pressure. Global economic output reached $100.56 trillion in 2021, with a population of 7.837 billion. The shift from agricultural to urban populations driven by industrialization and modernization has brought about environmental pollution, resource waste, and challenges in freshwater and energy supply, threatening sustainable development. In response, international efforts, such as the signing of agreements like the "World Environmental Convention" and the "United Nations Framework Convention on Climate Change," aim to minimize the negative externalities of

**Data Availability Statement:** All relevant data are within the manuscript and its Supporting Information files.

**Funding:** This research was funded by [Social Science Foundation project of Shaanxi Province] grant number [2022D035] and [Shaanxi Normal

University graduate pilot talent fund project] grant number [LHRCTS23013]. The funders had no role in study design, data collection and analysis, decision to publish, or preparation of the manuscript.

**Competing interests:** The authors have declared that no competing interests exist.

industrialization and modernization on the global environment. This collective action seeks to continuously improve ecological environments, enhance Earth's ecological livability, and achieve sustainable development.

Since the reform and opening-up in China, economic growth has been rapid. However, the accompanying large-scale urbanization has presented severe challenges, including environmental pollution and resource shortages. Improving urban living environments and enhancing the carrying capacity of the ecological environment have become urgent issues. In 2015, the State Council of China issued the "Guiding Opinions on Promoting Sponge City Construction," outlining the promotion of pilot projects for "sponge cities." Can this initiative reduce human activities' environmental pressure, effectively improve ecological livability, and what are its implementation pathways and characteristics? This manuscript attempts to answer these questions, providing a Chinese solution for raising Earth's ecological livability and achieving ecological sustainable development.

Academic research on "sponge cities" originated from the "Low Impact Development Theory" proposed in the 1980s in the United States. Subsequently, Australia and New Zealand conducted research on "Water Sensitive Urban Design" and "Low Impact Urban Design and Development" by drawing on the experience of stormwater management in the United States. In 2005, Huw Pohlner of the International Water Management Institute in India officially introduced the term "sponge city." In 2012, China further explained the concept of "sponge city" at the "2012 Low Carbon City and Regional Development Technology Forum." Since then, "sponge cities" have sparked enthusiastic discussions and close attention worldwide in both theoretical and practical circles. According to existing studies, the core concept of "sponge city" can be summarized as giving full play to the intervention functions of natural ecology and artificial ecology to minimize the impact of urban development and construction on the ecological environment (Wu Danjie et al., 2016) [1].The construction and development of "sponge cities" is of great significance in coping with natural disasters and climate change (Ma et al., 2023) [2], alleviating urban waterlogging and pollution problems (Yang et al., 2022) [3], and conserving water resources (Stephan et al., 2023) [4].

Ecological livability has been a hot topic in academic discussions, focusing on its conceptual definition, measurement levels, and influencing factors. In terms of conceptual definition, ecological livability originated in the early 21st century and has developed several viewpoints [5], including the human-oriented perspective, which considers ecological livability as development that satisfies residents' needs for a comfortable life while meeting ecological sustainable development requirements [6–8]. The co-development perspective regards ecological livability as the coordinated development of the economy, society, and ecology [9–11]. The integrated development perspective defines ecological livability as the integrated development of the natural ecological environment and the human and social environment [12]. Regarding measurement levels, Ye Qing et al. (2011) constructed a two-dimensional vector structure indicator system, evaluating ecological livability from both soft (behavioral processes) and hard (outcome effects) aspects [13]. Saitluanga (2014) and Zheng Chundong et al. (2014) measured ecological livability based on resident satisfaction, considering both objective and subjective indicators [14,15]. In terms of influencing factors, Yang et al. (2023) found that green finance played a key role in improving ecological livability [16]. Yue et al. (2023) considered green space as a crucial aspect of building ecologically livable cities [17].

As for whether the "sponge city" pilot can reduce the environmental pressure of human activities and effectively improve ecological livability, and what are its action paths and characteristics, Han Youting (2023) believes that "sponge city" can improve the ecological environment quality of the city by using the force of nature to drain water and reduce pollution [18]. Jingyu Wang et al. (2023) believe that the construction of "sponge city" can significantly

promote the improvement of ecological environment by solving urban waterlogging and non-point source pollution [19]. Ma Jing et al. (2023) used the system dynamics model to study and found that the construction of "sponge city" has achieved good results in the green and sustainable development of cities [20]. Yao Mingqi et al. (2023) believe that the pilot construction of "sponge city" significantly improves the ecological resilience of cities and is a sustainable urban development model [21]. In addition, Basnou et al. (2015) and Parker et al. (2019) discussed its impact on the ecological environment from the perspective of green infrastructure, and found that green infrastructure can effectively improve the ecological livability of cities [22,23].

The academic community has explored "sponge cities" and ecological livability, achieving fruitful research results. However, there is limited literature examining the impact of "sponge cities" on the level of ecological livability. Therefore, building upon mechanism analysis, this manuscript employs a multi-period difference-in-differences model to examine the impact of "sponge city" pilot projects on ecological livability. Potential innovations in this manuscript include: (1) constructing an indicator evaluation system for ecological livability from three dimensions—natural greenery level, residential comfort level and environmental governance level—and calculating the ecological livability of 288 cities in China using the entropy weight TOPSIS method; (2) incorporating "sponge city" pilot projects into the research perspective, expanding theoretical and empirical research on how "sponge city" pilot projects improve ecological livability; (3) dividing the research sample based on aridity levels and per capita water supply, clarifying the heterogeneity characteristics of "sponge city" pilot projects in improving ecological livability; (4) exploring the mediating effects of human capital aggregation and green technology innovation in the impact of "sponge city" pilot projects on ecological livability.

## Theoretical analysis and hypotheses

Ecological livability refers to a composite system that encompasses both a harmonious ecological environment, comfortable living conditions, and a civilized social environment. It reflects the sustainable development capacity of a given region. According to the fundamental tenets of ecological theory, human economic and social development should involve the rational and efficient development and utilization of resources. Simultaneously, it should leverage the self-regulation capacity of ecosystems to ensure mutual dependence of resources and system sustainability [24]. The "sponge city" pilot projects represent an organic integration of modern green technologies with various factors [25], including the environment, society, and humanities. Through innovative ecological planning and sustainable development principles, these projects aim to improve ecological livability by developing and utilizing resources in a rational and efficient manner.

From an ecological environment perspective, the "sponge city" pilot projects can effectively coordinate natural ecological functions and artificial intervention functions. During rainfall, they absorb, store, infiltrate, and purify water, releasing stored water when needed for utilization. This contributes to the restoration of urban aquatic ecosystems, reduces the occurrence of urban floods, and effectively enhances urban ecosystem functions and microclimate regulation.Regarding the residential environment, the construction elements of "sponge city" pilot projects, including sponge-type buildings and neighborhoods, sponge-type roads and squares, sponge-type parks and green spaces, urban drainage facilities, and water bodies, are closely related to residents' living conditions. These developments are beneficial for improving residents' water supply capabilities and optimizing their living conditions.From a social environment perspective, "sponge city" pilot projects can use technical measures such as ecological ponds and artificial wetlands to filter pollutants in rainwater. This prevents sudden pollution from surface runoff and sediment in the sewer network, which could lead to pollutant

concentrations exceeding normal sewage levels. This approach reduces runoff pollution, enhances pollution control levels, and improves ecological livability. Based on the above discussions, the following research hypotheses are proposed:

**Hypothesis 1:** "Sponge city" pilot projects can enhance ecological livability.

According to Schultz's human capital theory, migration is an investment in individual human capital aimed at enhancing economic benefits and overall quality of life [26]. The construction of "sponge city" pilot projects, through the integration of modern green technologies with social, environmental, and human factors, creates new job opportunities, improves the income levels and development opportunities of relevant employees, attracting a considerable influx of high-level human capital to migrate and cluster in pilot cities.

Simultaneously, a higher level of human capital agglomeration in a region is conducive to improving ecological livability. On the one hand, human capital agglomeration can generate a "learning effect" through communication and collaboration among workers in related industries, thereby expanding the channels for the dissemination of ecological and environmental knowledge and skills. This is advantageous for promoting the accumulation of ecological and environmental technologies and innovations within the region, indirectly enhancing ecological livability. On the other hand, the "spillover effect" generated by human capital agglomeration can expedite the diffusion of advanced ecological livability concepts. This, in turn, propels local governments to establish higher environmental standards, providing assurance for the enhancement of local ecological livability through rigorous environmental supervision and management policies. Therefore, this manuscript posits the second research hypothesis:

**Hypothesis 2:** "Sponge city" pilot projects enhance ecological livability by promoting human capital aggregation.

According to Mueser's technology innovation theory (R.Mueser, 1985), technological innovation is a process involving new ideas and discontinuous technological activities, which, over time, develop into practical and successful applications [27]. "Sponge city" pilot projects, through measures such as subsidizing research and development and utilizing public-private partnership (PPP) projects, promote the development and practical application of green engineering technology products, fostering green technological innovation.

Meanwhile, green technology innovation can effectively improve ecological livability. Firstly, green technology innovation can reduce energy consumption and emissions. Green technology innovation helps to improve industrial production equipment and processes, promote the transformation of high pollution production methods to eco-friendly production methods, enhance the intensity of production factors through intensification effects, improve resource utilization efficiency, achieve energy conservation and emission reduction, and thus help to enhance the ecological livability of the region. Secondly, green technology innovation can improve the level of ecological environment pollution control. Green technology innovation can not only achieve "source control" by upgrading environmental governance technology, that is, to prevent and control pollution and emissions in the process of pollutant manufacturing, but also improve the efficiency and level of end of pipe treatment, that is, to reduce the emissions of pollutants into the environment through technological treatment before the waste is discharged into nature, thereby improving the level of ecological livability. Based on the above discussion, this manuscript proposes a third research hypothesis:

**Hypothesis 3:** "Sponge city" pilot projects enhance ecological livability by promoting green technological innovation.

## Data and methodology

### Data source and variable selection

We collected data on "sponge city" pilot projects and ecological livability for 288 cities in China from 2010 to 2021. The choice of 2010 as the starting point for the study is justified for the following reasons: (1) The "sponge city" pilot policy in China commenced in 2015. To facilitate estimation, it is essential to retain samples from before policy implementation. (2) The sampling interval should not be too long, as longer intervals may introduce interference from other policies.

**Dependent variable.** Ecological livability should encompass human needs for greenery, health, and well-being. In this manuscript, an evaluation indicator system for ecological livability was constructed based on three aspects: the degree of natural greenery, residential comfort, and environmental governance. Ecological livability was then measured using the entropy weight TOPSIS method. Table 1 presents the evaluation indicator system for ecological livability. Data sources include the "China City Statistical Yearbook" and the "China Smart City Yearbook" from 2010 to 2021, as well as the "China Urban Construction Statistical Yearbook."

**Core explanatory variables.** In October 2015, the General Office of the State Council of China issued the "Guiding Opinions on Promoting the Construction of Sponge Cities," emphasizing the acceleration of sponge city construction to promote harmonious development between humans and nature. We consider the "Sponge City" pilot policy as a quasi-natural experiment and conducted a difference-in-differences estimation to assess the impact of this exogenous event. Specifically, within the sample period, there were 28 cities and two new districts designated as "Sponge City" pilots. Among them, Gui'an New Area is a new area formed by the combination of Guiyang City and Anshun City, and the area of Xi'an New Area involves two cities, Xi'an and Xianyang. To ensure the accuracy of the evaluation of the "Sponge City" pilot policy, we selected these 32 cities as the experimental group and the remaining 256 non-pilot cities as the control group. The data were sourced from the list of "Sponge City" pilot cities published by the Ministry of Finance, the Ministry of Housing and Urban-Rural Development, and the Ministry of Water Resources.

**Table 1. Evaluation indicator system for ecological livability.**

| Target Layer | System Layer | Criterion Layer | Symbol | Definition | Attribute |
|---|---|---|---|---|---|
| Ecological Livability | Natural Greenery Level | Built-up Area Green Space Ratio | Land | The ratio of the green space area within the built-up area to the total land area of the administrative region | (+) |
| | | Built-up Area Green Coverage Ratio | Green | The ratio of the green coverage area within the built-up area to the total land area of the administrative region | (+) |
| | Residential Comfort Level | Population Density | Population | The ratio of registered population to the total area of the city | (-) |
| | | Air Quality | PM2.5 | The average concentration of particulate matter that can be inhaled | (-) |
| | | Per Capita Water Supply | Water | The ratio of total water resources to the registered population | (+) |
| | Environmental Governance Level | Sewage Treatment Capacity | Sewage | Centralized treatment rate of wastewater treatment plants | (+) |
| | | Solid Waste Utilization | Solid | The rate at which general industrial solid waste is comprehensively utilized | (+) |
| | | Wastewater Discharge per Unit of Industrial Output Value | Effluent | The amount of industrial wastewater discharged relative to the total industrial output value | (-) |
| | | SO2 Emissions per Unit of Industrial Output Value | SO2 | The quantity of sulfur dioxide (SO2) emitted by industries in relation to the total industrial output value | (-) |
| | | Particulate Matter (PM) Emissions per Unit of Industrial Output Value | Smoke | The amount of particulate matter (PM) emitted by industries relative to the total industrial output value | (-) |

**Mechanism variables.**   Drawing on previous research [16], we selected human capital aggregation and green technology innovation as mechanism variables. Data on human capital aggregation are from the "China Urban Statistical Yearbook" for the years 2010–2021, and data on green technology innovation are sourced from the China National Intellectual Property Administration. Green patents were classified based on the WIPO green patent list, providing the total number of granted green patents.

**Control variables.**   Consistent with prior literature, we included the following control variables: technological level, economic level, foreign capital level, and financial scale. Data for control variables were derived from the "China Urban Statistical Yearbook" for the years 2010–2021 and the Wind database. Table 2 presents the specific definitions of the main variables.

### Descriptive statistics

Table 3 reports the descriptive statistics of the variables. The mean of Ecology is 0.06, indicating that the average level of ecological livability in the sample cities from 2010 to 2021 is 0.06. The minimum and maximum values of Ecology are 0.02 and 0.74, respectively. The maximum value is 12 times the mean and 37 times the minimum, suggesting that there is significant room for improvement in the ecological livability of most cities. All raw data in the table have been standardized in the subsequent econometric process. The standardization formula for positively oriented indicators is as follows:

$$Z_+ = \frac{X - X_{min}}{X_{max} - X_{min}} \tag{1}$$

For negatively oriented indicators, the standardization formula is:

$$Z_- = \frac{X_{max} - X}{X_{max} - X_{min}} \tag{2}$$

In the above, $Z_+$ and $Z_-$ respectively denote the positive and negative standardized values of the sample, $X_{min}$ represents the minimum value of the sample, and $X_{max}$ represents the maximum value of the sample.

**Table 2. Variable definitions.**

| Variables | Index | Symbol | Definition |
|---|---|---|---|
| Dependent Variable | Ecological Livability | Ecology | Refer to Table 1 |
| Core Explanatory Variables | "Sponge City" Pilot | DID | From the Ministry of Finance, the Ministry of Housing and Urban-Rural Development, the Ministry of Water Resources, the three ministries of the review panel reviewed the "sponge city" pilot city list |
| Mediating Variables | Human Capital Aggregation | Human | Number of employees in water, environmental, and public facilities management industry |
| | Green Technology Innovation | Innovate | Total number of green patents granted |
| Control Variables | Technological Level | Science | Number of research and development personnel |
| | Economic Level | Economy | Per capita regional gross domestic product |
| | Foreign Capital Level | Foreign | Foreign direct investment enterprise industrial output / (Domestic enterprise industrial output + Hong Kong, Macao, and Taiwan investment enterprise industrial output) |
| | Financial Scale | Finance | Per capita year-end balance of deposits in financial institutions |

Table 3. Variable definitions.

| Variable | N | Mean | SD | Min | Max |
|---|---|---|---|---|---|
| Land | 3456 | 0.72 | 1.67 | 0.00 | 20.39 |
| Green | 3456 | 0.80 | 1.87 | 0.00 | 21.97 |
| Population | 3456 | 4.32 | 3.42 | 0.05 | 26.48 |
| PM2.5 | 3456 | 47.98 | 26.54 | 10.00 | 347.00 |
| Water | 3456 | 20.37 | 27.37 | 0.29 | 442.30 |
| Sewage | 3456 | 87.30 | 13.40 | 10.38 | 121.90 |
| Solid | 3456 | 78.48 | 23.84 | 0.24 | 200.00 |
| Effluent | 3456 | 0.00 | 0.00 | 0.00 | 0.24 |
| SO2 | 3456 | 0.00 | 0.02 | 0.00 | 0.78 |
| Smoke | 3456 | 0.00 | 0.11 | 0.00 | 5.77 |
| Ecology | 3456 | 0.06 | 0.06 | 0.02 | 0.74 |
| DID | 3456 | 0.06 | 0.24 | 0.00 | 1.00 |
| Human | 3456 | 0.83 | 1.07 | 0.00 | 12.53 |
| Innovate | 3456 | 373.80 | 1096.00 | 0.00 | 18238.00 |
| Science | 3456 | 1.98 | 4.11 | 0.00 | 53.53 |
| Economy | 3456 | 5.33 | 3.43 | 0.53 | 46.77 |
| Foreign | 3456 | 0.15 | 1.97 | 0.00 | 108.61 |
| Finance | 3456 | 9.39 | 13.37 | 0.76 | 170.10 |

Note: This table shows the descriptive statistics. N is the number of samples. Mean is the mean of variables. SD is the standard deviation, Min is the minimum value. Max is the maximum value and P50 is the median.

## Regression model

The differential model (DID) can perform regression based on individual data to judge whether the influence of policies is statistically significant, and can avoid the endogeneity problem caused by policies as the core explanatory variable to a large extent. Since the time of the implementation of the "sponge city" pilot is inconsistent, and the traditional DID model can only evaluate the implementation effect of the policy at a single time node, the traditional DID model is not suitable for this manuscript. The multi-period Difference-in-Differences (DID) model can effectively deal with the situation where the implementation time of the policy is not completely consistent, and is suitable for the gradual implementation of the same policy in the sample group.The regression model is specified as follows, given the involvement of two phases of the "sponge city" pilot, and considering different policy periods, we employ a multi-period Difference-in-Differences (DID) model. The baseline regression model is set as follows:

$$\text{Ecology}_{it} = \alpha_0 + \alpha_1 \text{DID}_{it} + \alpha_2 X_{it} + \mu_i + \gamma_t + \varepsilon_{it} \tag{3}$$

I and t represent the city and year, respectively. $\text{Ecology}_{it}$ represents ecological livability. $\text{DID}_{it}$ is the core explanatory variable,takes a value of 1 if city i in year t is a "sponge city" pilot, and 0 otherwise. The coefficient of $\alpha_1$ indicates the marginal contribution of the sponge city pilot construction to ecological livability. A positive coefficient implies that the "sponge city" pilot has a positive impact on ecological livability $X_{it}$ represents a series of control variables. $\mu_i$ is the individual fixed effect. $\gamma_t$ is the time fixed effect. $\varepsilon_{it}$ is the random disturbance term.

## Empirical analysis

### Feature facts

The dynamic evolution characteristics of ecological livability in China mainly include several aspects (Fig 1). First, during the sample period from 2010 to 2021, the center of the kernel density estimate curve moves to the right, and the left tail gradually shorten, the right tail gradually lengthens, indicating a continuous improvement in China's ecological livability over time. Second, the distribution pattern of the peaks shows a "steep-flat" fluctuation trend. Compared with the years 2016 and 2021, the peak of the kernel density curve for China's ecological livability in 2010 is steeper. The peak of the kernel density estimate curve decreases sequentially over the sample period, indicating an increase in spatial disparity in China's ecological livability during this period. Third, the number of peaks changes insignificantly, showing a clear unimodal characteristic. This suggests that China's ecological livability did not exhibit significant polarization during the sample period.

### Parallel trend test

We employ the event study method to test for parallel trends, ensuring the validity of the Difference-in-Differences (DID) identification. Drawing on the methodology of Jacobson et al. (1993) [28], we construct the following econometric model:

$$\text{Ecology}_{it} = \alpha_0 + \sum_{k=-3}^{5} \beta_k \times D_{i,t+k} + \alpha_1 X_{it} + \mu_i + \gamma_t + \varepsilon_{it} \tag{4}$$

In the equation, $D_{i,t+k}$ is the dummy variable for 'sponge city' pilot, and in this section, we select the three years before policy implementation and the five years after policy implementation for parallel trend testing. $\beta_k$ represents the difference in ecological livability between the experimental group and the control group before and after the implementation of the 'sponge city' pilot policy.

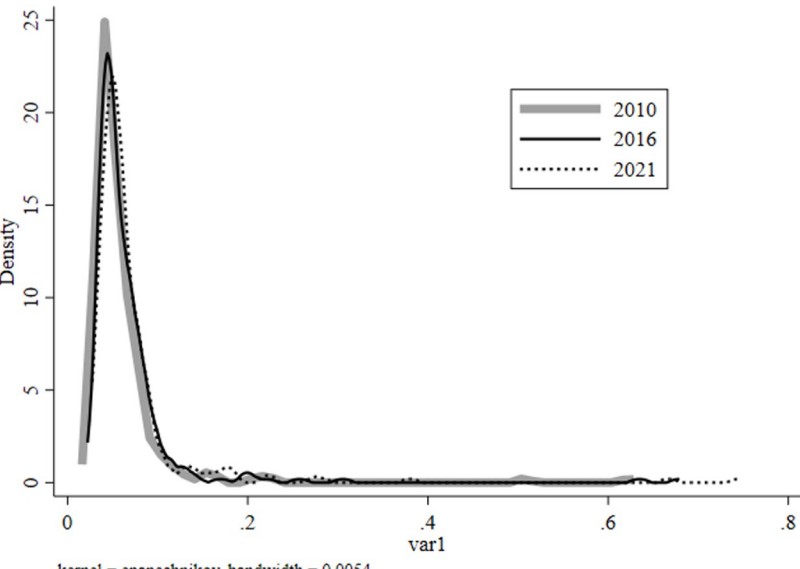

kernel = epanechnikov, bandwidth = 0.0054

**Fig 1. Kernel density estimation of ecological livability.**

Fig 2 presents the test results of estimated coefficients and 95% confidence intervals. We observe that the estimate of the pre-reform dummy variable is not significant, indicating the presence of a parallel trend in ecological livability between the treatment and control group cities before the implementation of the 'sponge city' pilot policy. We find that in the first year after the implementation of the 'sponge city' pilot policy, the estimate is not statistically significant. However, in the second year and beyond the policy implementation, the estimate becomes statistically significant and shows a positive trend, indicating a significant positive impact of the 'sponge city' pilot on ecological livability. Moreover, this boosting effect exhibits a certain duration of lagging effects.

## Benchmark regression

In order to explore the impact of the "sponge city" pilot policy on ecological livability, we used a multi-period Difference-in-Differences (DID) model for empirical test. Table 4 presents the baseline regression results of the impact of the 'sponge city' pilot on ecological livability. Model (1) interprets the ecological livability using the policy variable DID as the sole explanatory variable, controlling for time fixed effects and individual fixed effects. Models (2)-(5) gradually incorporate additional control variables on the basis of Model (1), and with the addition of control variables, the goodness of fit of the models slightly improves. Model (5) indicates that for every 1% enhancement in the 'sponge city' pilot, ecological livability increases by 0.003 percentage points. It shows that "sponge city" can improve ecological livability through innovative ecological planning and sustainable development concept, reasonable and efficient development and utilization of resources. Confirming Hypothesis 1. Additionally, improvements in technological level and financial scale significantly enhance ecological livability, while an increase in the level of foreign investment significantly diminishes ecological livability.

## Robustness testing

**Placebo test.**　To avoid the potential impact of unobservable omitted variables, we employ the placebo test by randomly assigning the treatment group through the bootstrapping method

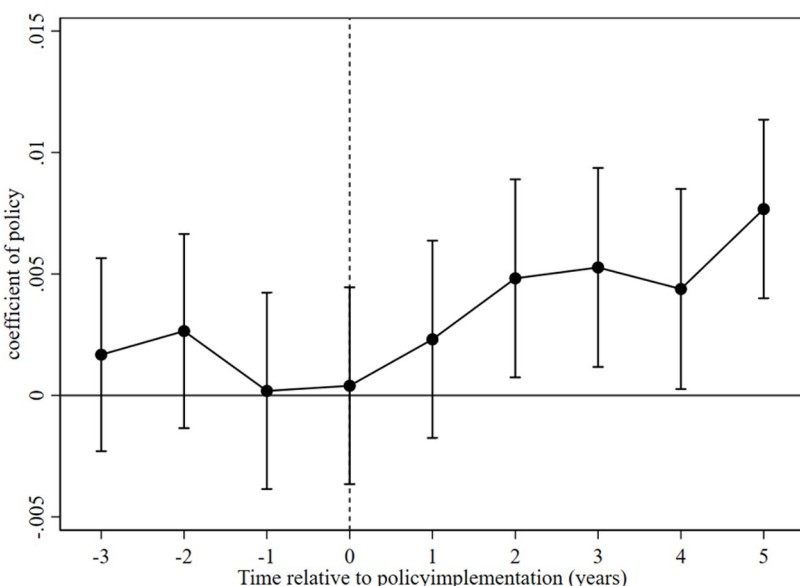

**Fig 2. Parallel trend test.**

**Table 4. The baseline regression results.**

| Variables | (1) | (2) | (3) | (4) | (5) |
|---|---|---|---|---|---|
| DID | 0.010*** | 0.008*** | 0.007*** | 0.007*** | 0.003*** |
| | (9.272) | (7.030) | (6.762) | (6.767) | (3.213) |
| Science | | 0.104*** | 0.099*** | 0.096*** | 0.025** |
| | | (10.187) | (9.502) | (9.113) | (2.287) |
| Economy | | | 0.020*** | 0.019*** | 0.004 |
| | | | (2.866) | (2.813) | (0.556) |
| Foreign | | | | -0.011* | -0.041*** |
| | | | | (-1.660) | (-6.395) |
| Finance | | | | | 0.140*** |
| | | | | | (16.458) |
| Constant | 0.064*** | 0.060*** | 0.059*** | 0.059*** | 0.057*** |
| | (355.852) | (149.278) | (77.219) | (73.934) | (73.803) |
| Time FE | Yes | Yes | Yes | Yes | Yes |
| Individual FE | Yes | Yes | Yes | Yes | Yes |
| R2 | 0.975 | 0.976 | 0.976 | 0.976 | 0.978 |
| N | 3456 | 3456 | 3456 | 3456 | 3456 |

Note: Numbers in parentheses indicate t-test

* denotes $P < 0.1$

** denotes $P < 0.05$

*** denotes $P < 0.01$. Unless otherwise specified, all subsequent tables are the same as this table.

and repeating the simulation 500 times. As shown in Fig 3, the estimated values from the 500 random simulations closely follow a normal distribution, with coefficient estimates mostly differing from those in the baseline regression. Therefore, the constructed instrumental variable DID for ecological livability has no statistically or economically significant impact, ensuring the robustness of the baseline regression results in this manuscript.

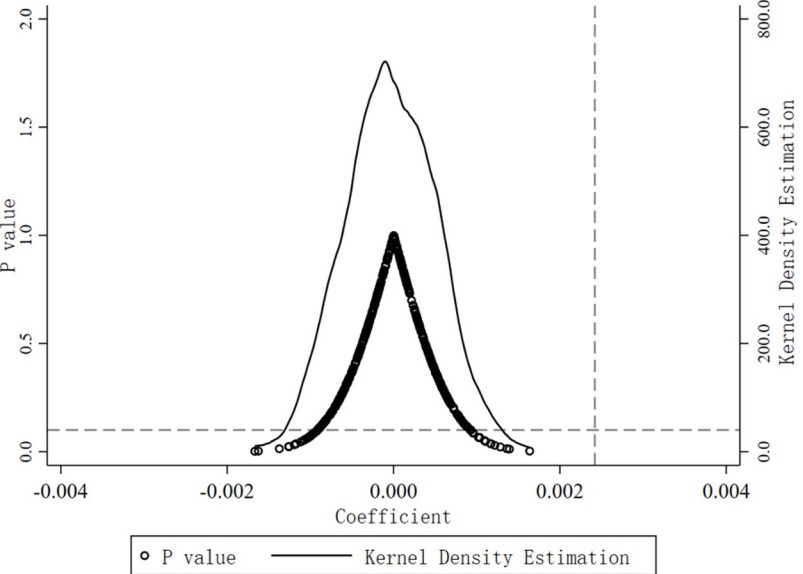

**Fig 3. Placebo test.**

**Replacing the dependent variable.**    The principal component analysis method was used to recalculate the dependent variable and perform a 1% level tail reduction. Model (1) in Table 5 shows the regression results of the "Sponge City" pilot on new ecological livability. The model controlled for both time fixed effects and individual fixed effects, but did not include control variables. The estimation results of model (1) show that the pilot project of "sponge cities" has a significant positive impact on ecological livability. Model (2) adds control variables to Model (1), and with the addition of control variables, the goodness of fit of the model slightly increases. The estimation results of model (2) show that for every 1% increase in the "sponge city" pilot, the ecological livability improves by 0.003 percentage points. This indicates that the "sponge city" pilot can effectively promote the improvement of ecological livability, which is consistent with the benchmark regression results, indicating that the benchmark regression model is robust.

**PSM-DID inspection.**    Due to the difficulty of overcoming sample selection bias in the DID method, we opted for the propensity score matching (PSM) method to process the sample, ensuring the effectiveness and accuracy of the regression results. Before conducting the PSM-DID test, we first performed an applicability test on the PSM-DID method, and the results are presented in Table 6. From Table 6, it can be observed that M1 and M2 represent the propensity score matching results obtained through caliper matching and kernel matching, respectively. Prior to matching, there were significant inter-group differences in the TEL, ECL, FIL, and FIS variables between the experimental and control groups. After matching, their T values significantly decreased, and the standard errors of TEL, ECL, and FIS variables were reduced to varying degrees. Therefore, the PSM method is effective for this study, and employing the PSM-DID method to examine the impact of "sponge city" pilot projects on ecological livability is appropriate.

Next, a regression analysis was conducted on the sample after propensity score matching. Table 7, columns (1) and (2), present the regression results for the samples obtained through caliper matching and kernel matching, respectively. It can be observed that the DID coefficients after matching remain significantly positive at the 1% level. This consistency with the baseline regression results from earlier sections further confirms that the "sponge city" pilot projects have a robust positive impact on ecological livability.

**Bilateral tail reduction processing.**    To explore whether outliers in the sample significantly affect the regression results, all continuous variables were subjected to two-sided trimming at the 1% and 5% levels. The regression was then conducted again, and the results are presented in Table 7, columns (3) and (4). The results indicate that the DID coefficients remain significantly positive at the 1% and 5% levels, suggesting that outliers in the sample

**Table 5.  Testing results for replacing the dependent variable.**

| Variables | (1) | (2) |
|---|---|---|
| DID | 0.003** | 0.003* |
|  | (2.782) | (2.875) |
| Constant | 0.567*** | 0.567*** |
|  | (3753.594) | (824.961) |
| Control Variable | No | Yes |
| Time FE | Yes | Yes |
| Individual FE | Yes | Yes |
| R2 | 0.778 | 0.779 |
| N | 3456 | 3456 |

**Table 6. Applicability testing of PSM-DID method.**

| Variable | Unmatched(U)/Matched (M) | Mean | | %bias | %reduct |bias| | t-test | | V(T)/V(C) |
|---|---|---|---|---|---|---|---|---|
| | | Treated | Control | | | t | p>|t| | |
| Science | U | 0.119 | 0.027 | 73.7 | - | 24.04 | 0.000 | 14.65* |
| | M1 | 0.072 | 0.073 | -0.6 | 99.2 | -0.11 | 0.909 | 1.28* |
| | M2 | 0.071 | 0.072 | -0.5 | 99.3 | -0.10 | 0.920 | 1.30* |
| Economy | U | 0.151 | 0.098 | 63.8 | - | 13.71 | 0.000 | 2.00* |
| | M1 | 0.137 | 0.140 | -4.2 | 93.4 | -0.55 | 0.584 | 0.72* |
| | M2 | 0.137 | 0.140 | -4.1 | 93.6 | -0.54 | 0.590 | 0.74* |
| Foreign | U | 0.016 | 0.027 | -30.1 | - | -4.45 | 0.000 | 0.17* |
| | M1 | 0.016 | 0.015 | 2.1 | 93.0 | 0.50 | 0.615 | 1.05 |
| | M2 | 0.016 | 0.016 | 1.5 | 95.2 | 0.34 | 0.737 | 0.94 |
| Finance | U | 0.126 | 0.042 | 67.7 | - | 21.05 | 0.000 | 10.89* |
| | M1 | 0.082 | 0.081 | 0.6 | 99.2 | 0.13 | 0.898 | 0.89 |
| | M2 | 0.082 | 0.081 | 0.5 | 99.2 | 0.12 | 0.904 | 0.59 |

Note: * if variance ratio outside [0.82; 1.22] for U and [0.81; 1.23] for M.

have a minor impact on the regression results. The baseline regression results presented earlier have passed the robustness check.

**Mechanism verification.** The above analysis concludes that the "sponge city" pilot has a positive impact on ecological livability. We attempt to further explore its specific implementation mechanisms. According to the descriptions in hypotheses 2 and 3, the "sponge city" pilot may enhance ecological livability through pathways such as human capital agglomeration and green technology innovation. Therefore, we further employ a mediation effects model to test the above mechanisms.

The basic equation set for the mediation effects test refers to the relevant study by Wen Zhonglin et al. (2014) [29] and is specified as follows:

$$Ecology_{it} = \alpha_0 + \alpha_1 DID + \alpha_2\ contral_{it} + \gamma_i + \varepsilon_{it} \tag{5}$$

$$Z_{it} = \alpha_0 + \alpha_1 DID + \alpha_2\ contral_{it} + \gamma_i + \delta_{it} \tag{6}$$

$$Ecology_{it} = \alpha_0 + \alpha_1 DID + \alpha_2 Z_{it} + \alpha_3\ contral_{it} + \gamma_i + \theta_{it} \tag{7}$$

**Table 7. Robustness check.**

| Variable | PSM-DID | | Bilateral tail reduction processing | |
|---|---|---|---|---|
| | (1) | (2) | (3) | (4) |
| DID | 0.004*** | 0.004*** | 0.005*** | 0.002** |
| | (3.880) | (3.691) | (5.326) | (2.461) |
| Constant | 0.052*** | 0.052*** | 0.055*** | 0.050*** |
| | (57.401) | (56.579) | (61.802) | (60.271) |
| Control Variable | Yes | Yes | Yes | Yes |
| Time FE | Yes | Yes | Yes | Yes |
| Individual FE | Yes | Yes | Yes | Yes |
| R2 | 0.960 | 0.957 | 0.957 | 0.938 |
| N | 3314 | 3243 | 3456 | 3456 |

Where Human and Innovate are the mediating variables, primarily encompassing human capital agglomeration and green technology innovation, and are random error terms. Eq (5) is used to estimate the total impact of the "sponge city" pilot on ecological livability. Eq (6) estimates the impact of the "sponge city" pilot on mediating variables, i.e., human capital agglomeration and green technology innovation. Eq (7) estimates the impact of the "sponge city" pilot on ecological livability after controlling for the effects of mediating variables. Regressions are performed in the above form, yielding the mediation effects test results shown in Table 10.

In Table 8, Models (1) and (2) present the test results for human capital agglomeration. In Model (1), the impact mechanism of human capital agglomeration is positive and significant, indicating that the "sponge city" pilot attracts more professionals in water resources, environmental management, and public facilities employment. In Model (2), the coefficients of DID and Human are also significantly positive, suggesting that the "sponge city" pilot can enhance ecological livability by promoting human capital agglomeration, thereby supporting Hypothesis 2.

Models (3) and (4) in Table 8 present the test results for green technology innovation. The results of Model (3) show that the impact mechanism of green technology innovation is positive and significant, indicating that the "sponge city" pilot significantly enhances the level of green technology innovation. In Model (4), the coefficients of DID and Innovate are also significantly positive, suggesting that the "sponge city" pilot can enhance ecological livability by promoting green technology innovation, supporting Hypothesis 3.

**Heterogeneity analysis.**   To investigate whether the impact of the "sponge city" pilot on ecological livability differs under varying levels of urban drought, we divided the 288 cities into two types based on the average precipitation from 2010 to 2021: drought-prone areas and non-drought-prone areas, each with 1728 samples. The regression results in Table 9, columns (1)-(2), reveal that the impact of the "sponge city" pilot on ecological livability in drought-prone areas is significantly positive at the 1% level. The regression coefficient of the policy variable is 0.008, indicating that the "sponge city" pilot can significantly enhance the ecological livability of drought-prone areas. However, the impact of the "sponge city" pilot on non-drought-prone areas is not significant. The potential reason is that drought-prone areas require extensive water storage and face greater challenges in addressing water resource issues, making

**Table 8. Mediation effect test.**

| Variable | (1) | (2) | (3) | (4) |
|---|---|---|---|---|
| | HCA | ER | GTI | ER |
| DID | 0.004* | 0.003*** | 0.007*** | 0.003*** |
| | (1.487) | (3.052) | (2.826) | (2.798) |
| Human | | 0.050*** | | |
| | | (6.943) | | |
| Innovate | | | | 0.064*** |
| | | | | (8.976) |
| Constant | 0.040*** | 0.055*** | -0.035*** | 0.059*** |
| | (21.087) | (67.165) | (-18.167) | (73.863) |
| Control Variable | Yes | Yes | Yes | Yes |
| Time FE | Yes | Yes | Yes | Yes |
| Individual FE | Yes | Yes | Yes | Yes |
| R2 | 0.936 | 0.978 | 0.867 | 0.978 |
| N | 3456 | 3456 | 3456 | 3456 |

them more dependent on water infrastructure construction. The "sponge city" pilot plays a complementary role, effectively improving the ecological livability of drought-prone areas.

We further explore whether the impact of the "sponge city" pilot on ecological livability varies under different levels of per capita water supply in urban areas. We divided the 288 cities into three types based on the mean per capita water supply from 2010 to 2021: water-deficient, moderately water-supplied, and water-abundant, each with 1152 samples. From the regression coefficients of the policy variable in models (1)-(3) in Table 10, it is evident that the impact of the "sponge city" pilot on the ecological livability of water-deficient cities is the most significant, with a coefficient of 0.006 and statistical significance at the 1% level. This indicates a pronounced positive effect of the "sponge city" pilot on the ecological livability of water-deficient cities. The impact coefficients for moderately water-supplied and water-abundant cities are positive, but the effects are not statistically significant. The reason may be that water-deficient cities place greater emphasis on the "sponge city" pilot policy, allocating more funds and resources, resulting in a more pronounced positive effect in these cities compared to moderately water-supplied and water-abundant cities.

## Conclusion and suggestions

The "sponge city" pilot plays an important role in improving ecological livability. Based on the theoretical analysis of the impact of the "sponge city" pilot on ecological livability, this paper empirically tests the research hypotheses by using the multi-period Difference-in-Differences (DID) model. The main research conclusions are as follows: First, the ecological livability of 288 cities in China increases from low to high with the change of time from 2010 to 2021, and the spatial gap of ecological livability increases. Second, the benchmark regression model shows that the estimated coefficient of DID is 0.003 and is significant at the 1% level, indicating that the "sponge city" pilot can significantly improve ecological livability. This research conclusion is still valid after a series of robustness tests such as placebo test, replacement of core explanatory variables, and psm-did test. In addition, the improvement of scientific and technological level and financial scale will also significantly improve ecological livability, while the improvement of foreign investment will significantly weaken ecological livability. Thirdly, the heterogeneity test shows that the impact of the "sponge city" pilot on ecological habitability is characterized by drought degree heterogeneity and per capita water supply heterogeneity. In terms of the degree of drought, the coefficient of DID in arid areas was 0.008, and it was significant at 1% level, indicating that the "sponge city" pilot has a significant improvement effect on the ecological habitability of arid areas, but the impact on the ecological habitability of non-arid areas is not significant. In terms of per capita water supply, the DID coefficient of cities

**Table 9. Heterogeneity testing of drought severity.**

| Variable | (1) | (2) |
|---|---|---|
| | Arid areas | Non arid areas |
| DID | 0.008*** | -0.001 |
| | 9.075 | (-0.359) |
| Constant | 0.044*** | 0.067*** |
| | (64.770) | (41.835) |
| Control Variable | Yes | Yes |
| Time FE | Yes | Yes |
| Individual FE | Yes | Yes |
| R2 | 0.954 | 0.979 |
| N | 1728 | 1728 |

**Table 10. Heterogeneity testing of per capita water supply.**

| Variable | (1) | (2) | (3) |
|---|---|---|---|
| | Cities lacking water supply | Cities with moderate water supply | Cities with abundant water supply |
| DID | 0.006*** | 0.001 | 0.003 |
| | (6.573) | (0.356) | (1.411) |
| Constant | 0.044*** | 0.059*** | 0.066*** |
| | (49.344) | (33.242) | (37.628) |
| Control Variable | Yes | Yes | Yes |
| Time FE | Yes | Yes | Yes |
| Individual FE | Yes | Yes | Yes |
| R2 | 0.979 | 0.989 | 0.751 |
| N | 1152 | 1152 | 1152 |

lacking water supply is 0.006, and is significant at 1% level, indicating that the "sponge city" pilot has a significant improvement effect on the ecological livability of cities lacking water supply, but has no significant impact on the ecological livability of cities with medium and abundant water supply. Fourth, the intermediary effect test shows that the regression coefficients of human capital concentration and green technology innovation are both positive and significant at the 1% level, indicating that the "sponge city" pilot can improve ecological livability through human capital concentration and green technology innovation.

Based on the above discussions, the following recommendations are proposed:First, implement "sponge city" construction guided by scientific planning to promote the improvement of ecological livability. Utilize modern technologies, including Geographic Information Systems (GIS) and intelligent sensing technologies, to comprehensively understand a city's water resources and vulnerabilities. This data-driven approach will enable the formulation of scientifically sound and systematic plans, allowing cities to utilize resources more efficiently and significantly enhance their ecological livability.Second, adopt a "tailored to local conditions" approach. Based on the geographical conditions, climate characteristics, and water resource distribution of different regions, strategically plan rainwater collection and flood retention areas. Tailoring the goals and plans of "sponge city" construction to local conditions ensures personalized planning that maximizes adaptation to the distinct features of each region. This approach facilitates the global promotion and implementation of the "sponge city" concept, effectively promoting the improvement of ecological livability in a flexible manner.Third, enhance the cultivation of talents in water resources, environmental, and related industries to drive eco-friendly sustainable development through technological innovation. Training professionals in water resources and environmental fields should not only focus on imparting fundamental theoretical knowledge but also emphasize practical skills and innovation. Establish laboratories and research centers closely tied to the forefront of technology, encourage student involvement in research projects, and use ecological and environmental technology accumulation and innovation as engines to propel eco-friendly sustainable development, thereby achieving an elevation in ecological livability.

## Supporting information

**S1 Data.**
(ZIP)

## Acknowledgments

The authors greatly appreciate the anonymous reviewers for their very valuable comments on this paper.

## Author Contributions

**Data curation:** Qi Wang.

**Funding acquisition:** Qi Wang.

**Methodology:** Qi Wang, Qinmei Wang, Xi Wang.

**Software:** Qi Wang, Xi Wang.

**Supervision:** Qinmei Wang.

**Writing – original draft:** Qi Wang, Qinmei Wang.

**Writing – review & editing:** Xi Wang.

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
