## [Decision Letter · Decision Letter 0]

24 Mar 2024

PONE-D-23-44189Can "Sponge City" Pilots Enhance Ecological Livability: Evidence from ChinaPLOS ONE

Dear Dr. Wang,

Thank you for submitting your manuscript to PLOS ONE. After careful consideration, we feel that it has merit but does not fully meet PLOS ONE’s publication criteria as it currently stands. Therefore, we invite you to submit a revised version of the manuscript that addresses the points raised during the review process.

Reviewer #1: While this article addresses important issues, there are many gaps and problems to consider it in its current form. I offer below a few comments and suggestions, which I hope will be useful to improve your work.

- Notwithstanding the manuscript's endeavor to delineate varied categorizations of academic research, it either parallels pre-existing literature or falls short of introducing profound novel insights. Scholarly manuscripts are esteemed for their novelty and seminal contributions. Redundancies with extant literature or the absence of innovative perspectives diminish its perceived value.

- The field of ecological research offers a variety of methodologies and strategies. Attempting to encapsulate or categorize these within a single manuscript is commendably ambitious but poses a quandary—the balance between comprehensiveness and depth. While the manuscript aims to provide a comprehensive perspective, it risks compromising depth, critical assessment, or intricate explication of specific methodologies. The hallmark of academic literature lies in its depth, rigorous analysis, and groundbreaking revelations. Overextension risks diluting these attributes, offering only superficial overviews without profound insights into specific subjects or methodologies.

- The research question is rather generic. The literature review does not aim to address a specific problem in the literature: "Can this initiative reduce human activities' environmental pressure, effectively improve ecological livability, and what are its implementation pathways and characteristics?"

- Some choices seem unjustified or arbitrary:

“The mainstream view holds that "sponge cities" are a development model for urban construction that relies on low-impact development to achieve ecological self-circulation and sustainable development.”

- The findings remain mostly generic and descriptive. Some interesting considerations are reported in the concluding section, but these are not adequately articulated and backed by data.

Reviewer #2: Research Importance and Contribution: Firstly, the manuscript's focus on the challenges of environmental pollution, water scarcity, and energy supply constraints, and its exploration of the impact of "sponge city" pilot projects on ecological livability, are recognized as having significant theoretical and practical importance. This study provides deep theoretical and empirical support for understanding the role of "sponge city" projects in enhancing ecological livability.

Methodology Evaluation: The adoption of a difference-in-differences (DiD) model, based on panel data from 288 Chinese cities over the period 2010-2021, to analyze the impact of "sponge city" pilot projects, is an appropriate methodological choice. This approach effectively identifies changes in ecological livability before and after the implementation of "sponge city" projects.

Research Findings and Interpretation: The findings that "sponge city" pilot projects significantly enhance ecological livability, with results remaining robust across different models and specifications, are acknowledged. Further mechanistic analysis revealing the aggregation of human capital and innovation in green technologies as key factors influencing improvements in ecological livability deepens the understanding of the mechanisms by which "sponge city" projects have an impact.

Heterogeneity Testing: The study also conducts heterogeneity tests, uncovering that "sponge city" pilot projects have a more pronounced effect on improving ecological livability in arid regions and cities facing water scarcity. These findings offer valuable guidance for future project location selection and resource allocation.

Recommendations: Finally, it is advised that the authors address the limitations of the study and suggest directions for future research. By incorporating these insights, the manuscript can further contribute to the body of knowledge on sustainable urban development and the efficacy of "sponge city" initiatives in improving ecological livability.

We look forward to receiving your revised manuscript.

Kind regards,

Saeid Norouzian-Maleki, Ph.D.

Academic Editor

PLOS ONE

 [This research was funded by [Social Science Foundation project of Shaanxi Province]grant number [2022D035] and [Shaanxi Normal University graduate pilot talent fund project] grant number [LHRCTS23013].].  

Reviewers' comments:

Reviewer's Responses to Questions

**Comments to the Author**

1. Is the manuscript technically sound, and do the data support the conclusions?

Reviewer #1: Partly

Reviewer #2: Yes

2. Has the statistical analysis been performed appropriately and rigorously? 

Reviewer #1: Yes

Reviewer #2: Yes

3. Have the authors made all data underlying the findings in their manuscript fully available?

Reviewer #1: No

Reviewer #2: Yes

4. Is the manuscript presented in an intelligible fashion and written in standard English?

Reviewer #1: No

Reviewer #2: Yes

5. Review Comments to the Author

Reviewer #1: While this article addresses important issues, there are many gaps and problems to consider it in its current form. I offer below a few comments and suggestions, which I hope will be useful to improve your work.

- Notwithstanding the manuscript's endeavor to delineate varied categorizations of academic research, it either parallels pre-existing literature or falls short of introducing profound novel insights. Scholarly manuscripts are esteemed for their novelty and seminal contributions. Redundancies with extant literature or the absence of innovative perspectives diminish its perceived value.

- The field of ecological research offers a variety of methodologies and strategies. Attempting to encapsulate or categorize these within a single manuscript is commendably ambitious but poses a quandary—the balance between comprehensiveness and depth. While the manuscript aims to provide a comprehensive perspective, it risks compromising depth, critical assessment, or intricate explication of specific methodologies. The hallmark of academic literature lies in its depth, rigorous analysis, and groundbreaking revelations. Overextension risks diluting these attributes, offering only superficial overviews without profound insights into specific subjects or methodologies.

- The research question is rather generic. The literature review does not aim to address a specific problem in the literature: "Can this initiative reduce human activities' environmental pressure, effectively improve ecological livability, and what are its implementation pathways and characteristics?"

- Some choices seem unjustified or arbitrary:

“The mainstream view holds that "sponge cities" are a development model for urban construction that relies on low-impact development to achieve ecological self-circulation and sustainable development.”

- The findings remain mostly generic and descriptive. Some interesting considerations are reported in the concluding section, but these are not adequately articulated and backed by data.

Reviewer #2: Research Importance and Contribution: Firstly, the manuscript's focus on the challenges of environmental pollution, water scarcity, and energy supply constraints, and its exploration of the impact of "sponge city" pilot projects on ecological livability, are recognized as having significant theoretical and practical importance. This study provides deep theoretical and empirical support for understanding the role of "sponge city" projects in enhancing ecological livability.

Methodology Evaluation: The adoption of a difference-in-differences (DiD) model, based on panel data from 288 Chinese cities over the period 2010-2021, to analyze the impact of "sponge city" pilot projects, is an appropriate methodological choice. This approach effectively identifies changes in ecological livability before and after the implementation of "sponge city" projects.

Research Findings and Interpretation: The findings that "sponge city" pilot projects significantly enhance ecological livability, with results remaining robust across different models and specifications, are acknowledged. Further mechanistic analysis revealing the aggregation of human capital and innovation in green technologies as key factors influencing improvements in ecological livability deepens the understanding of the mechanisms by which "sponge city" projects have an impact.

Heterogeneity Testing: The study also conducts heterogeneity tests, uncovering that "sponge city" pilot projects have a more pronounced effect on improving ecological livability in arid regions and cities facing water scarcity. These findings offer valuable guidance for future project location selection and resource allocation.

Recommendations: Finally, it is advised that the authors address the limitations of the study and suggest directions for future research. By incorporating these insights, the manuscript can further contribute to the body of knowledge on sustainable urban development and the efficacy of "sponge city" initiatives in improving ecological livability.

6. PLOS authors have the option to publish the peer review history of their article (what does this mean?). If published, this will include your full peer review and any attached files.

Reviewer #1: No

Reviewer #2: No

---

## [Author Response · Author response to Decision Letter 0]

4 Apr 2024

Thanks very much for the valuable suggestions of reviewers, we have made the following modifications and improvements to the manuscript according to the prompts:

Reviewer #1: While this article addresses important issues, there are many gaps and problems to consider it in its current form. I offer below a few comments and suggestions, which I hope will be useful to improve your work.

- Notwithstanding the manuscript's endeavor to delineate varied categorizations of academic research, it either parallels pre-existing literature or falls short of introducing profound novel insights. Scholarly manuscripts are esteemed for their novelty and seminal contributions. Redundancies with extant literature or the absence of innovative perspectives diminish its perceived value.

Response：Thank you very much for your suggestions on the innovation of the manuscript. According to the reviewer's prompts, we have carefully revised the manuscript, and highlighted the innovation of the article in the abstract and text. The revised manuscript emphasizes more on the impact path of the "sponge city" pilot on ecological livability, which increases the value of the manuscript. 

- The field of ecological research offers a variety of methodologies and strategies. Attempting to encapsulate or categorize these within a single manuscript is commendably ambitious but poses a quandary—the balance between comprehensiveness and depth. While the manuscript aims to provide a comprehensive perspective, it risks compromising depth, critical assessment, or intricate explication of specific methodologies. The hallmark of academic literature lies in its depth, rigorous analysis, and groundbreaking revelations. Overextension risks diluting these attributes, offering only superficial overviews without profound insights into specific subjects or methodologies.

Response：Thank you very much for your suggestions on balancing the research breadth and depth of the manuscript. According to the reviewer's guidance and suggestions, we have revised the manuscript. This manuscript starts from the "sponge city" pilot policy to explore the impact of this policy on ecological livability. Therefore, we adopt the multi-period Difference-in-Differences (DID) model for empirical testing. In the process of revising the manuscript, we focus on introducing and emphasizing the multi-period Difference-in-Differences (DID) model, analyzing the value and significance of the multi-period Difference-in-Differences (DID) model for this manuscript, and increasing the depth of the manuscript. 

- The research question is rather generic. The literature review does not aim to address a specific problem in the literature: "Can this initiative reduce human activities' environmental pressure, effectively improve ecological livability, and what are its implementation pathways and characteristics?"

Response：Thank you very much for your suggestions on the literature review. According to the reviewer's suggestions, we have thought about and discussed this part, and revised the literature review. We have added the literature review on "Can this initiative reduce the environmental pressure of human activities, effectively improve ecological livability, and what are its implementation pathways and characteristics?" , making the literature review more suitable for the research problem of this manuscript.

- Some choices seem unjustified or arbitrary:

“The mainstream view holds that "sponge cities" are a development model for urban construction that relies on low-impact development to achieve ecological self-circulation and sustainable development.”

Response：We are very grateful for the reviewers' suggestions on the modification of unreasonable remarks in the manuscript. We have carefully modified the unreasonable or arbitrary remarks, and changed "The mainstream view holds that "sponge cities" are a development model for urban construction that relies on low-impact development to achieve ecological self-circulation and sustainable development." to "Existing research shows that the core concept of "sponge city" can be summarized as giving full play to the natural ecological and artificial ecological intervention functions, and minimizing the impact of urban development and construction on the ecological environment."

- The findings remain mostly generic and descriptive. Some interesting considerations are reported in the concluding section, but these are not adequately articulated and backed by data.

Response：We are very grateful for the reviewers' suggestions on the modification of the research conclusion. According to the reviewers' suggestions, we have modified the research conclusion: first, we have increased the data support for the research conclusion; second, we have made more detailed explanations for the research conclusion.

Reviewer #2: Research Importance and Contribution: Firstly, the manuscript's focus on the challenges of environmental pollution, water scarcity, and energy supply constraints, and its exploration of the impact of "sponge city" pilot projects on ecological livability, are recognized as having significant theoretical and practical importance. This study provides deep theoretical and empirical support for understanding the role of "sponge city" projects in enhancing ecological livability.

Response：We are very grateful for the reviewers' recognition and suggestions on the research significance of the manuscript. We have further deepened and highlighted the research significance and innovation points of the manuscript. 

Methodology Evaluation: The adoption of a difference-in-differences (DiD) model, based on panel data from 288 Chinese cities over the period 2010-2021, to analyze the impact of "sponge city" pilot projects, is an appropriate methodological choice. This approach effectively identifies changes in ecological livability before and after the implementation of "sponge city" projects.

Response：We are very grateful for the reviewers' recognition and suggestions on the research method of the manuscript. We have made further detailed introduction to the multi-period Difference-in-Differences (DID) model used in the manuscript, demonstrated the important value and significance of the multi-period Difference-in-Differences (DID) model in the manuscript, and increased the depth of the manuscript.

Research Findings and Interpretation: The findings that "sponge city" pilot projects significantly enhance ecological livability, with results remaining robust across different models and specifications, are acknowledged. Further mechanistic analysis revealing the aggregation of human capital and innovation in green technologies as key factors influencing improvements in ecological livability deepens the understanding of the mechanisms by which "sponge city" projects have an impact.

Response：We are very grateful for the reviewers' recognition and suggestions on the research conclusion of the manuscript. According to the suggestions of the reviewers, we have revised the research conclusions of the manuscript, provided a more detailed description of the influencing mechanism in the abstract section, and added data support in the research conclusion section. 

Heterogeneity Testing: The study also conducts heterogeneity tests, uncovering that "sponge city" pilot projects have a more pronounced effect on improving ecological livability in arid regions and cities facing water scarcity. These findings offer valuable guidance for future project location selection and resource allocation.

Recommendations: Finally, it is advised that the authors address the limitations of the study and suggest directions for future research. By incorporating these insights, the manuscript can further contribute to the body of knowledge on sustainable urban development and the efficacy of "sponge city" initiatives in improving ecological livability.

Response：We are very grateful to the reviewers for their recognition and suggestions on the heterogeneity test and research conclusions. We have supplemented data support for the heterogeneity test and other contents in the conclusion section, making the conclusion more complete and scientific.

In addition, we have uploaded all the data and code used in the manuscript to “Supporting Information”.

---

## [Decision Letter · Decision Letter 1]

2 May 2024

PONE-D-23-44189R1Can "Sponge City" Pilots Enhance Ecological Livability: Evidence from ChinaPLOS ONE

Dear Dr. Wang,

Thank you for submitting your manuscript to PLOS ONE. After careful consideration, we feel that it has merit but does not fully meet PLOS ONE’s publication criteria as it currently stands. Therefore, we invite you to submit a revised version of the manuscript that addresses the points raised during the review process.

Reviewer #1: The authors have improved the quality of their manuscript after the first round of revisions and currently have the following issues:

Firstly, in the abstract section, further revision is recommended.

Secondly, Figure 1. Kernel density estimation of ecological livability. The picture is not clear enough. Please replace it.

We look forward to receiving your revised manuscript.

Kind regards,

Saeid Norouzian-Maleki, Ph.D.

Academic Editor

PLOS ONE

Journal Requirements:

Reviewers' comments:

Reviewer's Responses to Questions

**Comments to the Author**

1. If the authors have adequately addressed your comments raised in a previous round of review and you feel that this manuscript is now acceptable for publication, you may indicate that here to bypass the “Comments to the Author” section, enter your conflict of interest statement in the “Confidential to Editor” section, and submit your "Accept" recommendation.

Reviewer #1: All comments have been addressed

Reviewer #2: All comments have been addressed

2. Is the manuscript technically sound, and do the data support the conclusions?

Reviewer #1: Partly

Reviewer #2: Yes

3. Has the statistical analysis been performed appropriately and rigorously? 

Reviewer #1: Yes

Reviewer #2: Yes

4. Have the authors made all data underlying the findings in their manuscript fully available?

Reviewer #1: Yes

Reviewer #2: Yes

5. Is the manuscript presented in an intelligible fashion and written in standard English?

Reviewer #1: Yes

Reviewer #2: Yes

6. Review Comments to the Author

Reviewer #1: The authors have improved the quality of their manuscript after the first round of revisions and currently have the following issues:

Firstly, in the abstract section, further revision is recommended.

Secondly, Figure 1. Kernel density estimation of ecological livability. The picture is not clear enough. Please replace it.

Reviewer #2: Thank you very much for all the co-authors careful revisions. The manuscript has valuable implications in the context of sponge cities.

7. PLOS authors have the option to publish the peer review history of their article (what does this mean?). If published, this will include your full peer review and any attached files.

Reviewer #1: No

Reviewer #2: No

---

## [Author Response · Author response to Decision Letter 1]

7 May 2024

Thanks very much for the valuable suggestions of reviewers, we have made the following modifications and improvements to the manuscript according to the prompts:

Reviewer #1: 

Firstly, in the abstract section, further revision is recommended.

Response：Thank you very much for your suggestions on the abstract. Based on the suggestions, we have carefully discussed and revised the grammar and logic of the sentences in the manuscript.

Secondly, Figure 1. Kernel density estimation of ecological livability. The picture is not clear enough. Please replace it.

Response：Thank you very much for your suggestions on the Figure 1 of the manuscript. According to the reviewer's prompts, we have carefully revised the Figure 1. Due to the close distance between the lines and the difficulty in widening their distance,so we mainly adjusted the color and style of the lines to make the figure clearer.

In addition, we have uploaded all the data and code used in the manuscript to “Supporting Information”.

---

## [Editor Report · Decision Letter 2]

10 May 2024

Can "Sponge City" Pilots Enhance Ecological Livability: Evidence from China

PONE-D-23-44189R2

Dear Dr. Wang,

We’re pleased to inform you that your manuscript has been judged scientifically suitable for publication and will be formally accepted for publication once it meets all outstanding technical requirements.

Kind regards,

Saeid Norouzian-Maleki, Ph.D.

Academic Editor

PLOS ONE
---

## [Editor Report · Acceptance letter]

16 May 2024

PONE-D-23-44189R2 

PLOS ONE

Dear Dr. Wang, 

I'm pleased to inform you that your manuscript has been deemed suitable for publication in PLOS ONE. Congratulations! Your manuscript is now being handed over to our production team.

Kind regards, 

on behalf of

Dr. Saeid Norouzian-Maleki 

Academic Editor

PLOS ONE